# The Effects of Diatomite as an Additive on the Macroscopic Properties and Microstructure of Concrete

**DOI:** 10.3390/ma16051833

**Published:** 2023-02-23

**Authors:** Chunqing Li, Guoyu Li, Dun Chen, Kai Gao, Yapeng Cao, Yu Zhou, Yuncheng Mao, Shanzhi Fan, Liyun Tang, Hailiang Jia

**Affiliations:** 1State Key Laboratory of Frozen Soil Engineering, Northwest Institute of Eco-Environment and Resources, Chinese Academy of Sciences, Lanzhou 730030, China; 2University of Chinese Academy of Sciences, Beijing 100049, China; 3Da Xing’anling Observation and Research Station of Frozen-Ground Engineering and Environment, Northwest Institute of Eco-Environment and Resources, Chinese Academy of Sciences, Da Xing’anling 165100, China; 4School of Civil Engineering, Northwest Minzu University, Lanzhou 730030, China; 5Gansu Provincial Transportation Research Institute Group Co., Ltd., Lanzhou 730050, China; 6Architecture and Civil Engineering School, Xi’an University of Science and Technology, Xi’an 710054, China

**Keywords:** diatomite, hydration reaction, physical and mechanical properties, pore structure, performance, microstructure

## Abstract

Diatomite is a siliceous sedimentary rock containing amorphous silica, which can be used as a green mineral admixture to improve the properties of concrete. This study investigates the affecting mechanism of diatomite on concrete performance by macro and micro tests. The results indicate that diatomite can reduce the fluidity of concrete mixture and change its water absorption, compressive strength, resistance to chloride penetration (RCP), porosity, and microstructure. The low fluidity of concrete mixture containing diatomite can reduce workability. With increasing diatomite as partial replacement for cement in concrete, water absorption of concrete decreases before increasing, while compressive strength and RCP rise first and then drop. When diatomite is added to the cement at a content of 5% by weight, the concrete has the lowest water absorption and the highest compressive strength and RCP. Through the mercury intrusion porosimetry (MIP) test, we determined that the addition of 5% diatomite reduces the porosity of concrete from 12.68% to 10.82% and changes the proportion of pores with different sizes in concrete, the proportion of harmless and less harmful pores increases, and the proportion of harmful pores reduces. Based on the microstructure analysis, the SiO_2_ in diatomite can react with CH and produce C-S-H. C-S-H is responsible for developing concrete because it fills pores and cracks, forms a platy structure, and makes the concrete much denser, thereby improving its macroscopic performance and microstructure.

## 1. Introduction

Diatomite consists primarily of the siliceous remains of ancient diatom communities and other microorganisms, and its main chemical composition is SiO_2_, with small amounts of Al_2_O_3_, Fe_2_O_3_, CaO, MgO, and others [1,2]. Diatomite has many naturally ordered pore channels with a pore size of 1.7 to 800 nm, a porosity of 60% to 80%, a specific surface area of 10 to 80 m^2^/g, a high adsorption capacity (can absorb 1.5 to 4 times its mass of water), good thermal stability (melting point 1650–1750 °C), and high chemical stability [3,4]. Due to its superior physicochemical properties, diatomite is used as an abrasive, absorbent, water treatment agent, air purifier, filter aid, catalyst, insulation product, functional filler, and construction material [5,6,7,8,9,10,11,12,13]. In particular, diatomite has prospects for wide application as construction material.

Diatomite was first discovered in Germany in 1833. In the last 200 years, diatomite development, purification, processing, and application have considerably evolved, as has a new understanding of it. Diatomite resources are abundant and widely distributed, with the United States and China ranking first and second, respectively, in terms of reserves of diatomite. China has the second largest supply of diatomite in the world, but more than 80% of it is low quality. Utilization and expansion of low-quality diatomite resources are crucial. If diatomite can be widely used as a green building material, it will be of great significance. Concrete is the most widely used construction material in the world, and it is utilized in the construction of buildings, highways, bridges, tunnels, hydraulic engineering, and special structures. Large-large use of concrete has consumed many natural resources and negatively impacted the ecological environment. Particularly, the cement in concrete mixes poses the greatest threat to the natural environment and human health. Due to its high amorphous silica content, diatomite is considered a possible substitute for cement in cementitious composites. This type of silica can react with calcium hydroxide to form calcium silicate hydrates, which are responsible for developing concrete strength [14]. Therefore, in recent years, diatomite obtained through rough processing of diatom ore has been used as an alternative cementitious material to cement in concrete. Adding the proper proportion of diatomite to cement-based materials can improve their performance. The method is simple, environmentally friendly and highly recommended. Many studies have been conducted on diatomite as a green construction material. Diatomite as an admixture can change the workability, physical and mechanical properties, and durability of cement-based materials [15]. Fragoulis [16] and Yilmaz [17,18] investigated the water absorption of ground diatomite, and the results showed that the mixing water requirement of diatomite is greater than that of ordinary cement due to its unique pore structure. In most cases, adding diatomite without special treatment to cement paste, mortar or concrete reduces their fluidity and deteriorates their workability [19,20,21]. The influence of diatomite on the mechanical properties of concrete varies with its content, production process, and mixing method. Wu et al. [22] indicated that the compressive strength of porous concrete decreases with the increase in diatomite admixture. The addition of pre-wetted diatomite has no negative effect on the compressive strength of high-strength concrete but slightly improves the splitting tensile strength and the tension–compression ratio [23]. The optimal amount of activated diatomite in concrete is 6%, and its compressive strength increased by 158% [24]. When the diatomite calcination temperature reaches 650 °C, the content is 2%, the volcanic activity achieves its maximum value, and the strength of high-performance concrete increases with the calcination time within 2 h [25]. Diatomite can also be added to the concrete after being mixed with other minerals. Diatomite and fly ash are mixed uniformly and ground as concrete admixture. With a decrease in the particle size of the mixture, the compressive strength and flexural strength of concrete at the early-stage increases; later, the strength rises first and then falls [26]. Ergün [27] used diatomite and marble powder instead of cement to make concrete specimens and demonstrated that the compressive strength of the specimens mixed with diatomite is higher than that of ordinary concrete specimens. Aydin et al. [28] illustrated that as the diatomite ratio increases in concrete, the compressive strength and modulus of elasticity decrease and increase abruptly. In addition, diatomite also has a positive impact on the durability of concrete. The incorporation of 20% diatomite reduces the porosity of concrete, improving the durability of reinforced concrete in nitric acid solution [29]. In addition, the incorporation of 15% diatomite reduces the internal stress of concrete, thereby enhancing its frost resistance [30]. Adding diatomite and pumice to pervious concrete raises the permeability of the concrete and drops its thermal conductivity [31].

Although numerous studies on the use of diatomite as a mineral admixture have been conducted, the following gaps in knowledge require further investigation. First, most analyses have focused primarily on the effect of diatomite on the physical and mechanical properties of cement paste and mortar, while the effect of concrete properties has received less attention. In addition, the results of studies on the effects of diatomite on concrete properties are diverse. Second, diatomite is employed to replace cement in concrete as a mineral additive, and there are various studies on its replacement of cement mass below 2% or above 10%, but almost none between 3% and 9%. Furthermore, the results of comprehensive studies on the effects of diatomite on the workability, physical and mechanical properties and durability of concrete are limited, especially due to the absence of multi-scale research methods combining macroscopic and microscopic. This research addresses the above deficiencies and provides theoretical support for applying diatomite in concrete. Based on the eco-balance evaluation method of European standards EN ISO 14040 and EN ISO 14044 [32,33], the widespread use of diatomite instead of cement in concrete will benefit the ecological environment and improve the quality of human life. Therefore, it is essential to study the improvement of concrete performance with diatomite as a green additive.

## 2. Experiments

### 2.1. Materials

Ordinary Portland cement (P.O 42.5), conforming to the Chinese standard GB175-2007 [34], was used when preparing the concrete samples in the study. The used cement has a surface area of 345 m^2^/kg. Table 1 shows a chemical component of cement. Natural river sand (2.69 g/cm^3^, 1.0% mut content) and crushed stone (2.85 g/cm^3^, 0.6% mut content) were utilized based on the Chinese standard [35]. Polycarboxylate superplasticizer with binder materials of 1.0 wt.% was applied to modify the concrete’s fluidity. The diatomite employed in the experiment was sourced from diatom ore processed in the Changbai area of Jilin Province.

Figure 1 depicts the preparation process of diatomite. Diatom ore was transformed into white powder diatomite through the process of crushing, separation, calcination, grinding and sieving at Linjiang Shengmai diatomite Functional materials co., Ltd. Figure 2 illustrates the dissolution of the prepared diatomite in water. Diatomite was not easily soluble in water because it precipitated after being evenly stirred and left to stand for 10 min. Figure 3 reveals the microstructure of diatomite by scanning electron microscopy. The tiny particles of diatomite are shaped like sunflower disks and have numerous pores arranged in a regular pattern on their surface. X-ray fluorescence spectrometer was utilized to examine the chemical composition of the diatomite (Table 1 and Figure 4). Diatomite consists primarily of SiO_2_ and small amounts of Al_2_O_3_, Fe_2_O_3_, K_2_O, CaO, TiO_2_, MgO, and others. Table 2 lists the main physical parameters of diatomite.

### 2.2. Mix Proportion and Sample Preparation

#### 2.2.1. Mix Proportion

Table 3 lists the concrete mix proportion. A constant water/binder ratio of 0.4 was used for all the test samples. The variables were diatomite contents in concrete. The concrete samples without diatomite addition were named OPC as the control samples. The concrete samples created by replacing 3%, 5%, and 7% diatomite for cement were designated DC3, DC5, and DC7, respectively.

#### 2.2.2. Preparation of Samples

The size of concrete samples was 100 × 100 × 100 mm^3^ for compressive strength and water absorption tests (20 groups). The size of concrete samples was ϕ100 × 50 mm^3^ for the coulomb electric flux test (4 groups). The size of mortar samples was ϕ8 × 10 mm^3^ for the mercury intrusion porosimetry test (4 groups). The size of cement paste samples with water/binder ratio of 0.4 was 40 × 40 × 160 mm^3^ for the thermogravimetry and Fourier transform infrared spectroscopy tests (4 groups).

#### 2.2.3. Curing Conditions

Various types of concrete samples were cast in different molds. The produced samples were demolded after 24 h and cured in the standard curing room until testing. The room temperature was 20 ± 2 °C, and the relative humidity was greater than 95%.

### 2.3. Test Procedures

#### 2.3.1. Slump

The slump value was measured after uniformly stirring the concrete mixture of each mix proportion. Based on the Chinese standard GB/T 50080-2016 [36], the concrete mixture was evenly distributed in three layers within the slump cylinder. Each layer of the concrete mixture was evenly tamped 25 times with a tamper. The slump value was measured with a steel ruler once the slump time reached the 30 s.

#### 2.3.2. Water Absorption

Referring to ASTM C 642 standard [37] and according to the Chinese standard GB/T 50081-2019 [38], three cube specimens with a side length of 100 mm were employed to test the water absorption of concrete. The test age for concrete water absorption was 28 days. The specimens were soaked in water for 48 h at 20 ± 2 °C before being removed and weighed, and then placed in a blast drying oven for 48 h at 105 ± 5 °C and then taken out and weighed. The absorption is expressed as follows:$$W=\frac{M_{s}-M_{d}}{M_{d}}\times 100\%,$$
where *M_s_* is the mass of the samples saturated with water, and *M_d_* is the dry mass of the samples.

#### 2.3.3. Compressive Strength

Compressive strength was evaluated on the concrete samples at 3, 7, 14, and 28 days. The concrete specimens of 100 × 100 × 100 mm^3^ cubes were investigated according to the Chinese standard GB/T 50081-2019 [38]. Three specimens of each mix proportion were tested under strain control procedures. During the test, continuous and uniform loading at a rate of 0.5 MPa/s was applied. The strength values measured with concrete specimens were all multiplied by a size conversion factor of 0.95.

#### 2.3.4. Coulomb Electric Flux

Testing the Coulomb electric flux adopted ϕ100 × 50 mm^3^ cylindrical specimens in accordance with the Chinese standard GB/T 50082-2009 [39] with the age of 28 days. Before the experiment, vacuum and saturated water conditions were created, and a 60 ± 0.1 V DC constant voltage was applied. The current was recorded every 5 min for 6 h after the power was turned on.

#### 2.3.5. Mercury Intrusion Porosimetry (MIP)

The concrete mixture was first filtered through a 4.75 mm sieve to remove the gravel and then poured into a disposable paper cup to produce mortar samples. After 28 days of curing, the samples of a size of ϕ8 × 10 mm^3^ were extracted from the mortar samples using an electric drill as the samples for the MIP test (Figure 5). The samples were placed in ethanol to prevent a hydration reaction before the test.

#### 2.3.6. Thermogravimetry

Cuboid cement paste samples measuring 40 × 40 × 160 mm^3^ were ground into powder and used for TG investigation. The mass of samples was determined as a function of temperature at a programmed one. The thermogravimetric (TG) and derivative thermogravimetric (DTG) curves were obtained. The TG analysis was conducted using a German-made TGA 209F3 analyzer with a uniform heating rate of 10 °C/min from room temperature to 1010 °C in a gas flow atmosphere.

#### 2.3.7. Fourier Transform Infrared Spectroscopy (FTIR)

Cement paste powders specimen was utilized for the FTIR test. The sample was placed in the detector and scanned from 4000 to 400 cm^−1^ at a resolution of 4 cm^−1^. The sample could absorb infrared of specific frequencies, causing the intensity of the interference light to change and be received by the detector, thereby obtaining the interference patterns of various samples.

#### 2.3.8. Scanning Electron Microscope (SEM)

Small fragments of concrete specimens crushed in the compressive strength test were utilized for the SEM test. The samples were dried at ambient temperature, then placed in a gold spraying equipment to spray gold on a paste–aggregate interface, and finally placed in an SEM instrument to scan and observe the microstructure (Figure 6 and Figure 7).

## 3. Results and Discussion

### 3.1. Physical and Mechanical Properties

#### 3.1.1. Slump and Water Absorption

Figure 8 depicts the slump values of concrete samples OPC, DC3, DC5, and DC7. The slump value of the concrete mixture decreases gradually with increasing diatomite content, indicating that diatomite addition weakens its workability. Due to the pore structure of diatomite particles (Figure 3), it can absorb free water from the concrete mixture. The greater the diatomite content, the less flowable the concrete mixture. Water absorption is an essential characteristic of concrete, as it is one of the most significant indicators of concrete’s durability. Figure 9 represents the water absorption of concrete samples OPC, DC3, DC5, and DC7; their water absorption is 4.60%, 4.71%, 4.28%, and 4.55%, respectively. These values indicate the presence of permeable voids in the concrete. The water absorption of concrete samples first reduces and then rises with an increase in the diatomite content. The water absorption of sample DC5 is the lowest, 0.32% lower than sample OPC. This demonstrates that adding diatomite to concrete affects its open porosity and compactness. A sufficient amount of diatomite can fill the pores in the concrete and make its structure denser, whereas an excessive amount of diatomite can cause concrete to dry and shrink, thereby creating more pores or voids and loosening its structure. Concrete with 5% diatomite content has the lowest open porosity, densest structure, and relatively good workability.

#### 3.1.2. Compressive Strength at Different Curing Ages

The compressive strength of concrete samples OPC, DC3, DC5, and DC7 at 3, 7, 14, and 28 days were tested, and the results are shown in Figure 10 and Table 4. During the same curing period, the compressive strength of the concrete samples initially increases and then decreases with increasing diatomite content as a replacement for cement in concrete. When the diatomite content is 5%, the compressive strength of the concrete sample reaches the highest. The compressive strength of DC3 is higher than that of OPC, but lower than that of DC5. In contrast, DC7 has a significantly lower compressive strength than OPC, DC3, or DC5. This indicates that when the diatomite content is less than 5%, the compressive strength of concrete can be improved, whereas when it is more than 5%, the compressive strength of concrete will be noticeably reduced. At three days of age, DC3 and DC5 have A^a^ values of 75.9% and 77.6%, respectively, which are higher than OPC with 70.6%. At 28 days of age, the compressive strengths of DC3 and DC5 are only 0.4 and 0.5 MPa greater than that of OPC. This demonstrates that adding 3% and 5% diatomite to concrete can accelerate the hydration reaction and raise its early compressive strength more effectively while having little effect on its late compressive strength.

The main factors of the Influence of diatomite on the compressive strength of concrete samples are the following: (1) The filler effect: The diatomite’s particles are relatively small, and they are uniformly mixed in the concrete mixture, which can fill the pores larger than the diameter’s particles, reducing the porosity of the concrete and improving the compactness. (2) The dilution effect: As an additive, diatomite replaces part of the cement in the concrete mixture, reducing cement’s content and the total hydration reaction of cement. (3) The pozzolanic effect: The main content of diatomite is amorphous silica, which can react with the hydration products in concrete to produce calcium silicate hydrate with cementation ability, and further improve the concrete’s performance.

The factors mentioned above have beneficial and detrimental effects on concrete. Filling pores and generating C-S-H gel are advantageous features. As diatomite content rises, diatomite will agglomerate and drop the hydration reaction, which is undesirable. The detrimental effects are insignificant when the diatomite content is less than 5%. The detrimental impacts are great when the diatomite content exceeds 5%. The coupling of favorable and unfavorable factors improves the compressive strength of concrete when the diatomite content is 5%. Hence, the compressive strength of concrete varies dynamically with the content of diatomite.

### 3.2. Resistance of Concrete to Chloride Penetration

The resistance to chloride penetration (RCP) of concrete was tested by Coulomb electric flux test. Figure 11 illustrates that the charge passed of samples OPC, DC3, DC5, and DC7 is 1867, 1811, 1614, and 1996, respectively. The smaller the charge passed, the stronger the RCP ability. With the increase in diatomite, the RCP ability of concrete initially increases but then weakens. When the diatomite content is 5%, the RCP ability of concrete is the strongest. The penetration and migration of chloride ions in concrete can be reduced when the content of diatomite is less than 5%, indicating that the concrete structure becomes compact. When the diatomite content exceeds 5%, the compactness of the concrete structure decreases. DC5 has the densest structure and the strongest RCP capability. DC3 and DC5 have superior RCP abilities compared to OPC, whereas DC7 has significantly inferior RCP capabilities. This shows that excessive diatomite additions can seriously deteriorate the compactness of concrete and reduce its durability. Excessive diatomite in concrete causes it to dry and shrink, resulting in many pores and voids inside and outside.

Figure 12 displays the relationship between the charge passed and the compressive strength of concrete samples. It indicates from the nonlinear fitting curve that the compressive strength of concrete decreases with the increase in the charge passed. Conversely, the higher the compressive strength of concrete, the smaller the charge passed and the stronger the RCP ability, demonstrating better durability. Therefore, the compressive strength of concrete can be utilized to predict its RCP ability, i.e., durability.

### 3.3. Effect of Diatomite on the Pore Structure of Concrete

Based on the above experiments, when the content of diatomite is 5%, the concrete’s physical mechanics and RCP ability were optimal. Therefore, the MIP test analyzed the difference in pore characteristics between DC5 and OPC. Figure 13 exhibits the pore size distribution of DC5 and OPC. The pore size distribution ranges between 7 nm and 10,000 nm. Pores of different sizes are classified as harmless pores (<20 nm), less harmful pores (20–50 nm), harmful pores (50–200 nm), and more harmful pores (>200 nm) [40]. The most probable aperture is the pore size corresponding to the peak on the pore size differential distribution curve. The most probable apertures of DC5 and OPC are 30.4 and 40.8 nm, respectively, with DC5 being smaller than OPC. As can be seen, diatomite mainly changes the pores of concrete smaller than 200 nm and reduces the most probable apertures.

Figure 14 depicts the pore proportions of samples. The proportion of harmless and less harmful pores in DC5 is larger than in OPC, while the proportion of harmful pores in DC5 is smaller than in OPC. This indicates that 5% content of diatomite can change the proportion of pores with different pore sizes in concrete, reducing harmful pores, increasing harmless and less harmful pores. The previous section has shown that the performance of DC5 is better than that of OPC. This also confirms that pore characteristics are one of the factors affecting concrete performance. The porosity of DC5 and OPC is 10.82% and 12.68%, respectively. The porosity of DC5 is lower than OPC. In general, the 5% content diatomite decreases the total porosity of concrete by reducing the most probable aperture and increasing the proportion of micropores (<50 nm), thereby improving the performance of concrete.

### 3.4. Effect of Diatomite on the Microstructure of Concrete

#### 3.4.1. TG Analysis

Figure 15 illustrates the TG and DTG curves of samples OPC and DC5. In the TG curves, the weight of the samples decreases as the decomposition temperature rises. Three endothermic peaks appear in the DTG curves. The chemical products can be identified based on endothermic peaks [41]. Three typical endothermic peaks are located between 30 and 290 °C, 400 and 470 °C, and 530 and 690 °C. These three temperature stages correspond to the dehydration and decomposition of calcium silicate hydrate (C-S-H), dehydration and decomposition of calcium hydroxide (CH), and decomposition of calcite (CaCO_3_), respectively [42,43,44,45]. Dehydration and decomposition of C-S-H result in a weight loss of 10.07% for OPC and 10.45% for DC5, with DC5 experiencing a greater weight loss than OPC. Similarly, the weight loss of OPC and DC5 due to calcite decomposition is 3.25% and 2.90%, respectively, with DC5 having a smaller weight less than OPC. However, dehydration and decomposition of CH cause OPC and DC5 to lose 5.42% and 5.31% of their weight, respectively, with OPC losing more weight than DC5.

The results indicate that both OPC and DC5 contain the compound products C-S-H, CH, and CaCO_3_. There are more CaCO_3_ and CH in OPC than in DC5. This means that diatomite consumes the CH generated by the hydration reaction after being added to the concrete. DC5 contains more C-S-H than OPC, showing that diatomite can induce more C-S-H gel in concrete. C-S-H gel fills the pores and improves compactness. In short, diatomite can increase the performance of concrete by participating in the cement hydration reaction.

#### 3.4.2. FTIR Analysis

Figure 16 presents the FTIR spectra of the studied concrete samples OPC and DC5. Based on Wu et al. [46], the infrared absorption band can be used to identify the physical phase in the FTIR spectra. The characteristic wavenumber 464 cm^−1^ corresponds to the absorption band generated by SiO_2_. The absorption band of DC5 is stronger than that of OPC, suggesting more SiO_2_ in DC5 than in OPC. This is reasonable because DC5 contains 5% diatomite, but OPC contains none, and the main component of diatomite is SiO_2_. The strong absorption band at 923 cm^−1^ is caused by the vibration of SiO_4_^2−^ groups. This absorption is more prominent for DC5 than OPC, indicating that the former contains more SiO_4_^2−^ groups. The characteristic wavenumbers 875, 1642, and 3642 cm^−1^ are induced by the vibration of ettringite (AFt), which possess approximately the same absorption bands, indicating the presence of nearly equal amounts of ettringite in OPC and DC5. The vibration of C-S-H causes the absorption bonds at 1423 and 3441 cm^−1^ wavenumbers. The absorption band of DC5 is stronger than that of OPC, demonstrating that DC5 contains more C-S-H. Accordingly, the addition of diatomite to concrete produces more C-S-H, which can reduce concrete porosity and improve its properties.

#### 3.4.3. SEM Analysis

In order to investigate the effect of diatomite on the microstructure of concrete, SEM testing was conducted on concrete samples crushed by the compressive strength test. Figure 17 and Figure 18 depict the microstructure of the OPC and DC5 samples at magnifications of ×500, ×5000, and ×30,000. It demonstrates that there are fewer cracks in DC5 than in OPC, indicating that DC5 has a denser structure than OPC. This is primarily attributable to the pozzolanic effect of diatomite, which can react with the hydration product CH to produce additional C-S-H with cementation ability. The high magnification micrographs show that OPC contains many crystal particles and dense platy structures in DC5. With the help of element analysis of Energy Dispersive Spectrometer equipment, it can be determined that the crystal particles are CH, and the platy structures are C-S-H. It is common knowledge that CH is detrimental to the performance of concrete, while C-S-H fills the pores and cracks in concrete to form a very dense platy structure, which has good integrity and high density and helps improve the concrete’s performance. This perfectly agrees with the TG and FTIR analysis, indicating less CH and more C-S-H in DC5 than in OPC.

## 4. Conclusions

Diatomite is a mineral silica source material rich in active SiO_2_, which can be added to the concrete as a green mineral mixture. The physical and mechanical properties, RCP ability, porosity, and microstructure of concrete containing diatomite are investigated, and the following conclusions are drawn:

(1) Diatomite can reduce the fluidity of concrete, leading to poor workability. It can also change the water absorption and compressive strength of concrete. With the addition of 5% diatomite, the concrete has the smallest water absorption and the greatest compressive strength.

(2) The charge passed of concrete decreases and then increases with the addition of diatomite. The smaller the charge passed, the stronger the RCP ability. The RCP ability is at its peak when diatomite content is 5%, resulting in the greatest durability. The compressive strength of concrete and RCP ability are positively correlated.

(3) The 5% diatomite is highly effective at reducing concrete’s porosity and enhancing its pore characteristic. The porosity of concrete can be decreased from 12.68% to 10.82%. The most probable apertures can be dropped from 40.8 to 30.4 nm. The proportion of harmless and less harmful pores increases, and the proportion of harmful pores falls.

(4) Compared to OPC, the studied DC5 with the pozzolanic effective has better concrete properties. More CH is contained in OPC than in DC5, which is harmful to concrete. More C-S-H gels are produced in DC5 than in OPC, which is beneficial to concrete. C-S-H gels effectively fill the concrete pores and cracks, optimize the internal structure, and improve the physical and mechanical properties and durability.

## Figures and Tables

**Figure 1 materials-16-01833-f001:**
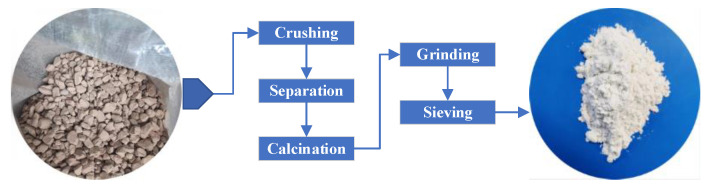
Preparation process of diatomite.

**Figure 2 materials-16-01833-f002:**
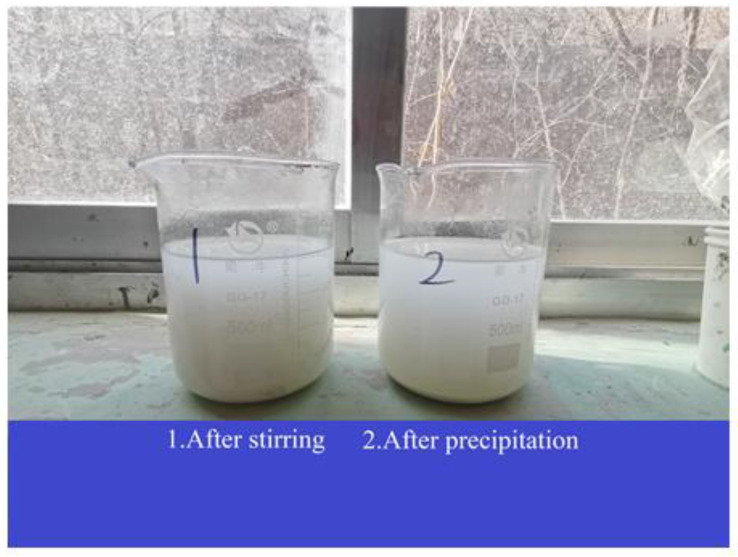
Diatomite suspension.

**Figure 3 materials-16-01833-f003:**
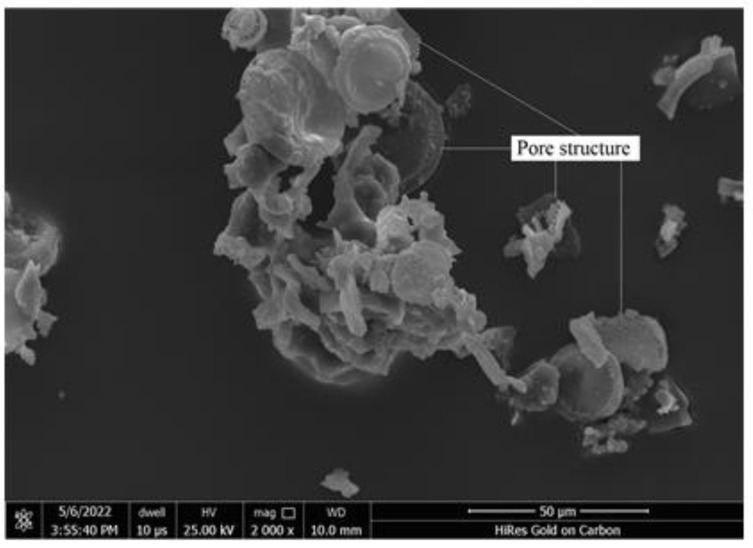
Microstructure of diatomite (×2000).

**Figure 4 materials-16-01833-f004:**
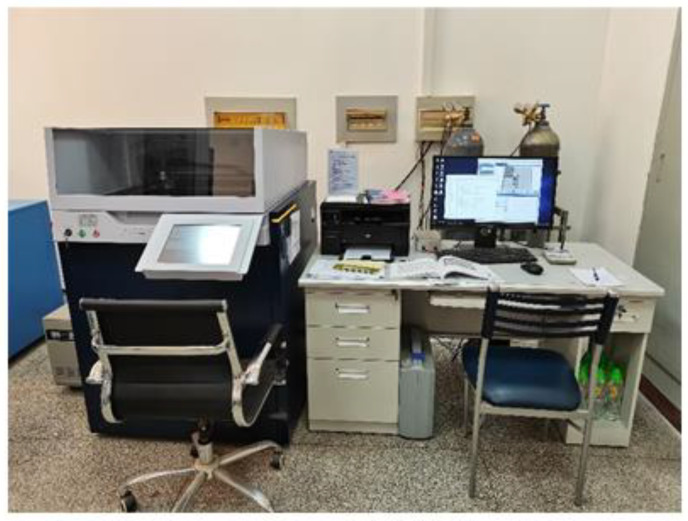
XRF equipment.

**Figure 5 materials-16-01833-f005:**
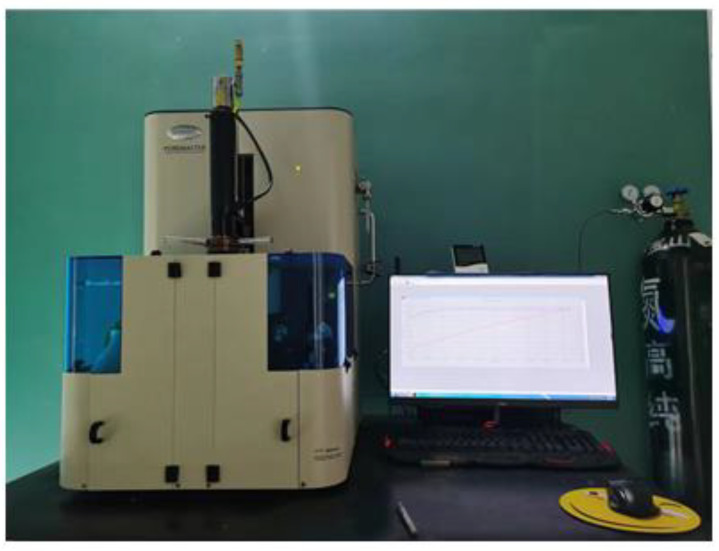
MIP equipment.

**Figure 6 materials-16-01833-f006:**
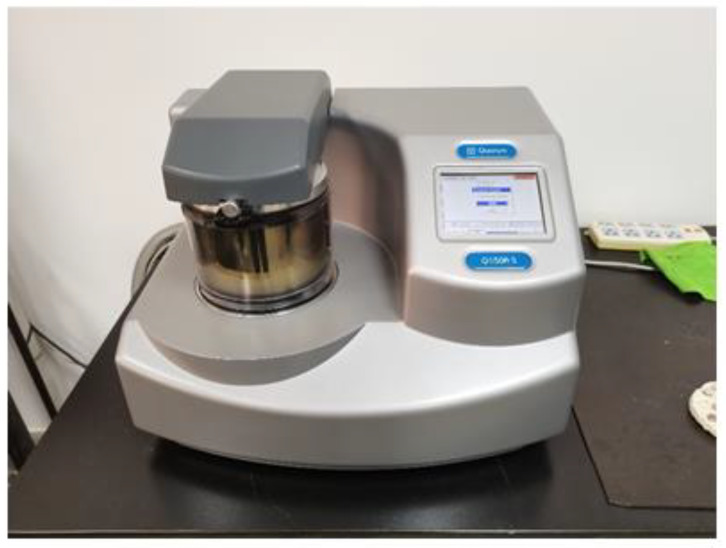
Gold spraying equipment.

**Figure 7 materials-16-01833-f007:**
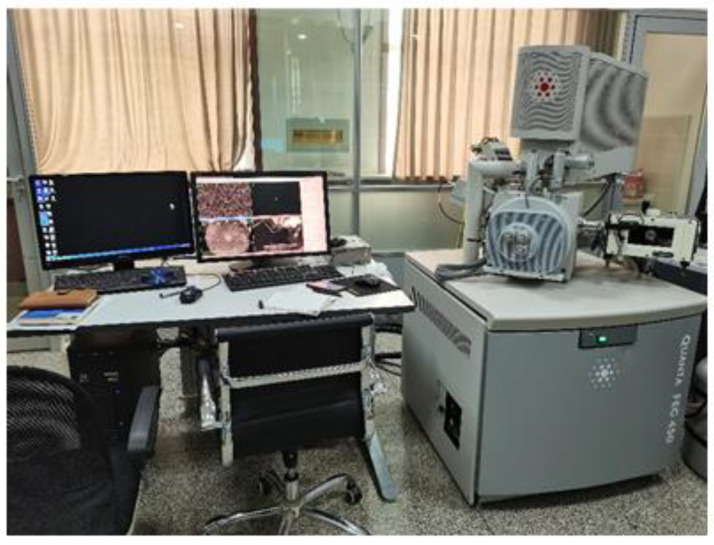
SEM equipment.

**Figure 8 materials-16-01833-f008:**
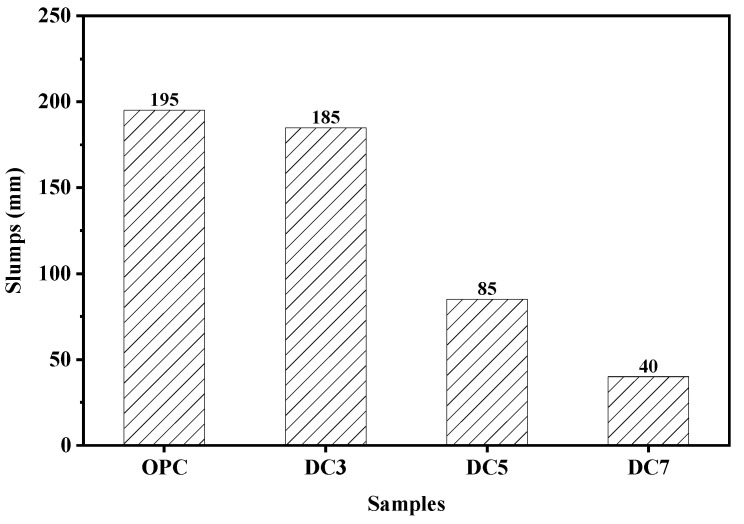
Slumps of samples.

**Figure 9 materials-16-01833-f009:**
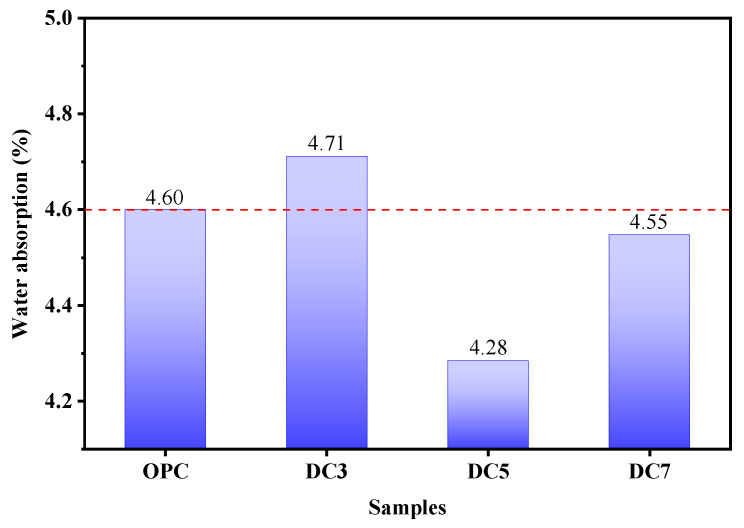
Water absorption of samples.

**Figure 10 materials-16-01833-f010:**
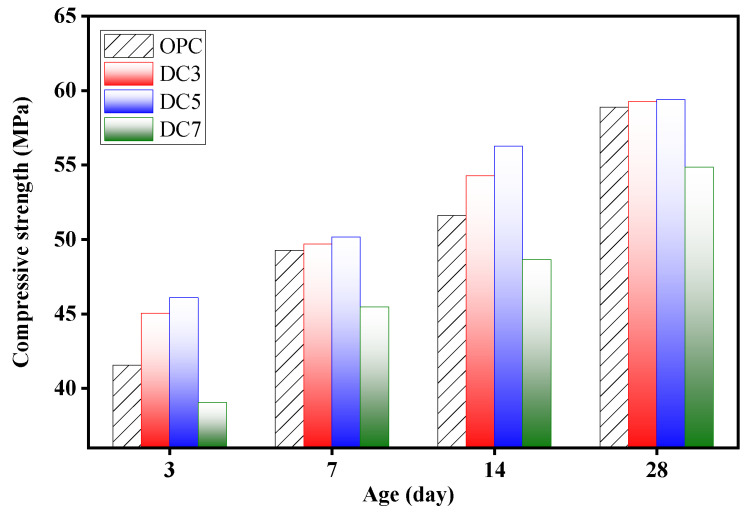
Compressive strengths of samples.

**Figure 11 materials-16-01833-f011:**
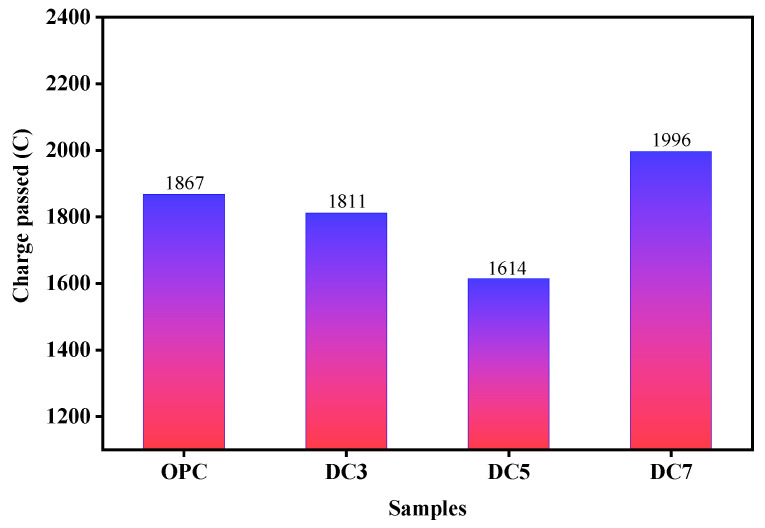
Charge passed of samples.

**Figure 12 materials-16-01833-f012:**
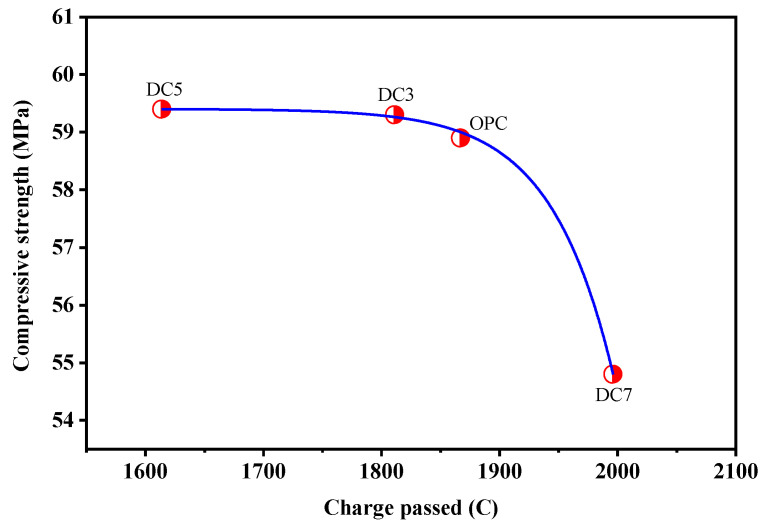
Relationship between charge passed and compressive strengths of concrete samples.

**Figure 13 materials-16-01833-f013:**
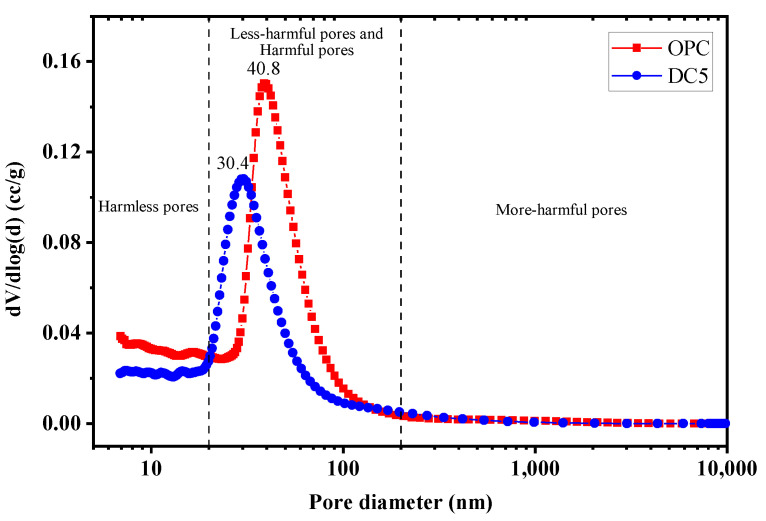
Pore size distribution of samples.

**Figure 14 materials-16-01833-f014:**
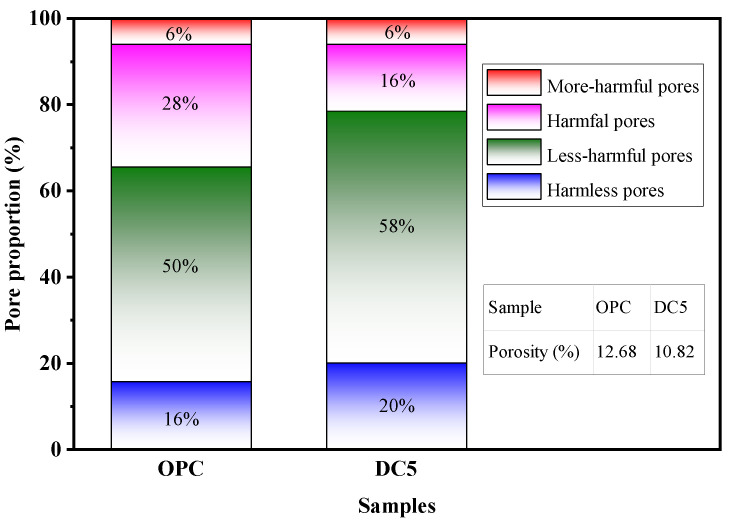
Pore proportions of samples.

**Figure 15 materials-16-01833-f015:**
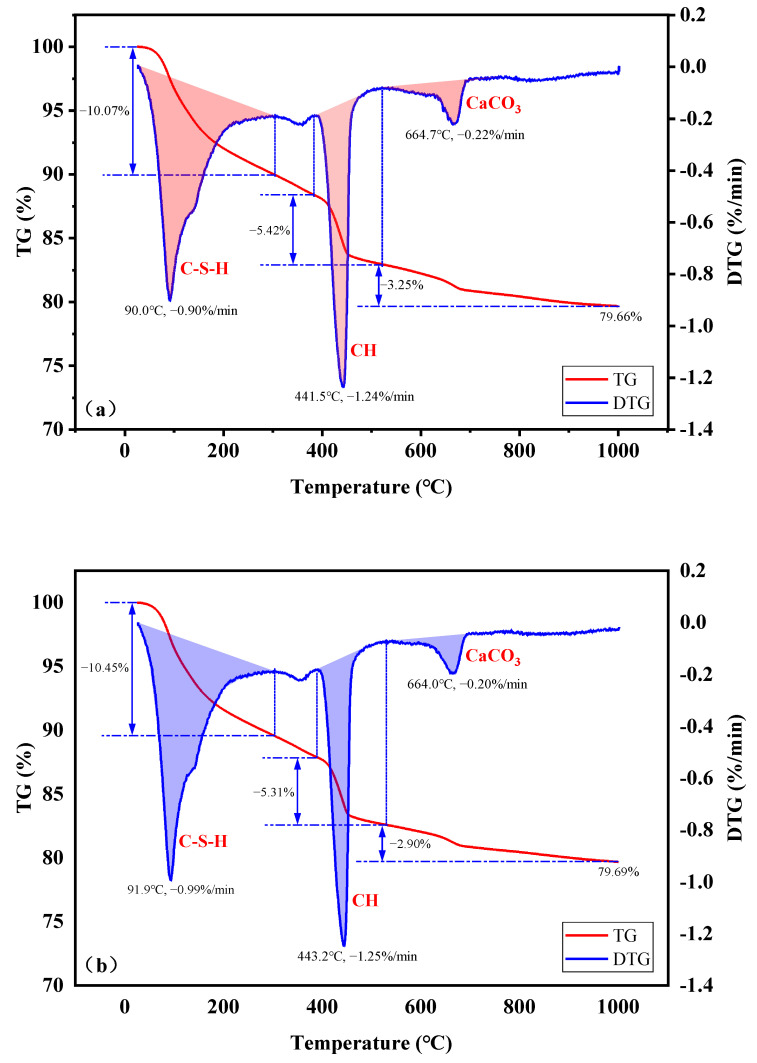
TG and DTG curves of samples: (**a**) OPC; (**b**) DC5.

**Figure 16 materials-16-01833-f016:**
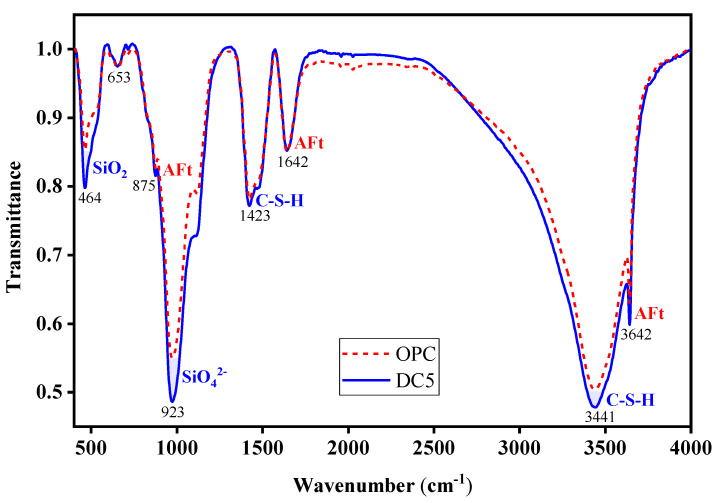
FTIR spectra of samples.

**Figure 17 materials-16-01833-f017:**
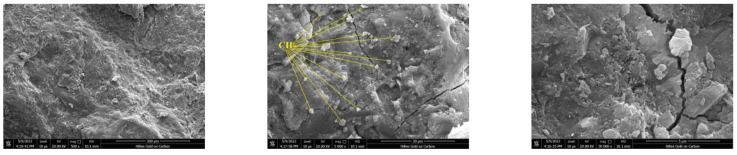
SEM micrographs of OPC (×500; ×5000; ×30,000).

**Figure 18 materials-16-01833-f018:**
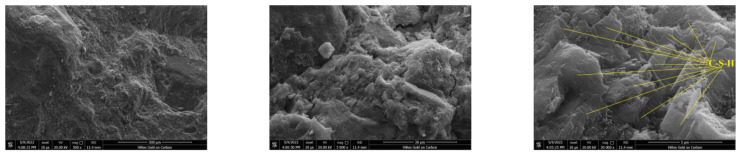
SEM micrographs of DC5 (×500; ×5000; ×30,000).

**Table 1 materials-16-01833-t001:** Chemical component of cement and diatomite (% by mass).

Chemical Material	CaO	SiO_2_	Al_2_O_3_	Fe_2_O_3_	MgO	K_2_O	TiO_2_	SO_3_	Others
Cement	69.89	17.30	3.14	3.76	2.09	0.64	0.18	2.25	0.75
Diatomite	0.31	89.31	1.68	1.50	0.15	0.40	0.17	0.05	1.45

**Table 2 materials-16-01833-t002:** Main parameters of diatomite.

Particle Size (μm)	SiO_2_ Content (%)	Permeability	Ignition Loss (%)	Bulk Density (g/cm^3^)	PH
<100	89.3	3.0	0.3	0.3–0.4	9.8

**Table 3 materials-16-01833-t003:** Mix proportion design (1 m^3^ concrete).

Sample	Water (kg)	Cement (kg)	Diatomite (kg)	Sand (kg)	Aggregate (kg)	Water Reducer (kg)	Production (Group)
OPC	176	440.0	0.0	678.0	1106.0	4.4	6
DC3	176	426.8	13.2	678.0	1106.0	4.4	6
DC5	176	418.0	22.0	678.0	1106.0	4.4	6
DC7	176	409.2	30.8	678.0	1106.0	4.4	6

**Table 4 materials-16-01833-t004:** Compressive strength values and rates of variation in compressive strength of the concretes.

CODE		Compressive Strength (MPa)					Variation vs. OPC (MPa)			
		3	7	14	28	A^a^	3	7	14	28
1	OPC	41.6	49.3	51.6	58.9	70.6	-	-	-	-
2	DC3	45.0	49.7	54.3	59.3	75.9	3.4	0.4	2.7	0.4
3	DC5	46.1	50.2	56.3	59.4	77.6	4.5	0.9	4.7	0.5
4	DC7	39.0	45.5	48.6	54.8	71.2	−2.6	−3.8	−3.0	−4.1

A^a^: Ratio of 3-day compressive strength to 28-day compressive strength of concrete (%).

## Data Availability

Data is contained within the article.

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
