# Peer review of "The Effects of Diatomite as an Additive on the Macroscopic Properties and Microstructure of Concrete"

_materials, 2023, doi:10.3390/ma16051833_

Round 1

Reviewer 1 Report

The authors have investigated the use of a Diatomite, a siliceous sedimentary rock, as a mineral admixture in concrete. While the paper would be able to capture the interest of researchers as it discusses on a new material in cementitious composites, the authors should address the following before the article can be considered for acceptance. 

1. In using Diatomite in cementitious composite, does it play a pozzolanic effect, latent hydraulic or filler effect. Please discuss this in detail. 

2. Was any test on reactivity of Diatomite's cementitious properties undertaken? Please discuss on the reactivity aspect of this material.

3. This statement in the Abstract: "However, the proportion of harmless and less harmful pores increases, and the proportion of harm-ful pores reduces". Can this statement be made prior to detailed durability analysis? Please justify and discuss. 

4. Abstract: "C-S-H is responsible for the development of concrete because it fills pores and cracks, forms a platy structure,...". Is the platy structured material C-S-H or calcium hydroxide? How was this determined? 

5. Was ultrasonication technique required in stirring the material to produce the Diatomite solution? 

6. Please discuss on the codal requirement on using this material in cementitious composites, with reference to at least the GB, ASTM and EN codes. 

Author Response

Thank you very much for your sincere suggestions, please see the attachment.

Reviewer 2 Report

The manuscript provides a great insight regarding the effect of DE as additive on the properties of concrete. However, it does require major revision in order to be considered for publication.

First and foremost, the manuscript shall be checked by native speaker.

The manuscript would be improved if in the introduction part the nature of diatomite is correlated with its use in this paper.

The manuscript requires restructuring. The materials and methods sections should be brief. The details should be placed in the results and discussion.

I truly appreciate the spirit of the author, however please consider removing unnecessary pictures (Fig 4 – Fig 7).

In the materials section (Fig. 1.) the author is discussing the sieving in the preparation process of diatomite. Why was sieving conducted? Or perhaps you needed the whole amount to have the same fractions. Please explain.

Table 3 discusses the mix design. Can you please elaborate the amount of samples tested for (compressive strength etc.)?

Did the author perform EDX on the samples? How did he conclude that the particles in Fig 17 are CH?

Did the authors perform XRPD? If yes, please include them. If not, please disregard this comment.

Author Response

(The authors gave the same response as above.)

Round 2

Reviewer 2 Report

I want to thank the authors for the great improvement of this manuscript.

My recommendation is that the manuscript is accepted for publication.